# The Clinical Chameleon of Autoinflammatory Diseases in Children

**DOI:** 10.3390/cells11142231

**Published:** 2022-07-18

**Authors:** Eugenio Sangiorgi, Donato Rigante

**Affiliations:** 1Department of Life Sciences and Public Health, Fondazione Policlinico Universitario A. Gemelli IRCCS, 00168 Rome, Italy; eugenio.sangiorgi@unicatt.it; 2Periodic Fevers Research Center, Università Cattolica Sacro Cuore, 00168 Rome, Italy

**Keywords:** autoinflammation, autoinflammatory disorder, periodic fever, interleukin-1, PFAPA syndrome, personalized medicine, child

## Abstract

The very first line of defense in humans is innate immunity, serving as a critical strongpoint in the regulation of inflammation. Abnormalities of the innate immunity machinery make up a motley group of rare diseases, named ‘autoinflammatory’, which are caused by mutations in genes involved in different immune pathways. Self-limited inflammatory bouts involving skin, serosal membranes, joints, gut and other districts of the human body burst and recur with variable periodicity in most autoinflammatory diseases (ADs), often leading to secondary amyloidosis as a long-term complication. Dysregulated inflammasome activity, overproduction of interleukin (IL)-1 or other IL-1-related cytokines and delayed shutdown of inflammation are pivotal keys in the majority of ADs. The recent progress of cellular biology has clarified many molecular mechanisms behind monogenic ADs, such as familial Mediterranean fever, tumor necrosis factor receptor-associated periodic syndrome (or ‘autosomal dominant familial periodic fever’), cryopyrin-associated periodic syndrome, mevalonate kinase deficiency, hereditary pyogenic diseases, idiopathic granulomatous diseases and defects of the ubiquitin-proteasome pathway. A long-lasting history of recurrent fevers should require the ruling out of chronic infections and malignancies before considering ADs in children. Little is known about the potential origin of polygenic ADs, in which sterile cytokine-mediated inflammation results from the activation of the innate immunity network, without familial recurrency, such as periodic fever/aphthous stomatitis/pharyngitis/cervical adenopathy (PFAPA) syndrome. The puzzle of febrile attacks recurring over time with chameleonic multi-inflammatory symptoms in children demands the inspection of the mixture of clinical data, inflammation parameters in the different disease phases, assessment of therapeutic efficacy of a handful of drugs such as corticosteroids, colchicine or IL-1 antagonists, and genotype analysis to exclude or confirm a monogenic origin.

## 1. Prelude to the Concept of Autoinflammation

Innate immunity exploits biochemical weapons in different tissues of the human body and has its main alarm system switch in the “inflammasome,” a multiprotein complex made of pattern recognition receptors and caspase-1, which processes pro-interleukin (IL)-1 into its mature form, IL-1β. All activated innate immune cells display their antibacterial activity through the inflammasome-mediated production of powerful inflammatory cytokines, such as IL-1β, to counteract microbial threats [1,2]. The discovery of monogenic defects in the innate immunity has clarified that both inflammasome activity and IL-1 release are altered in a cluster of rare diseases marked by recurrent inflammatory symptoms affecting skin, serosal membranes, joints, gut, central nervous system and other tissues, in which IL-1β represents the main driver of inflammation [3,4]. The word “autoinflammatory” in ‘autoinflammatory diseases’ (ADs) describes the “apparent” spontaneous burst of inflammation, as infectious triggers, autoreactive T lymphocytes and/or specific autoantibodies are absent [5]. ADs are caused by impaired production of pro-inflammatory cytokines, with IL-1β playing a seminal role, along with a delayed shut-down of the immune response, leading to recurrent episodes of fever, and in some cases, inflammation limited to specific organs [6]. These diseases can be categorized in hereditary monogenic and multifactorial polygenic disorders encompassing many protean conditions such as periodic fever/aphthous stomatitis/pharyngitis/cervical adenopathy (PFAPA) syndrome, which is still the most mysterious cause of idiopathic recurrent fevers in childhood [7]. All these disorders represent a diagnostic challenge because their symptoms are nonspecific and sometimes refer to infectious or malignant diseases. Although many classification criteria have been developed to help diagnose rare monogenic ADs, patients with recurrent fevers and inflammatory symptoms experience a general delay in identification of the disease, which can lead to further morbidities and dreadful complications, such as AA-amyloidosis [8,9]. The repertoire of ADs was expanded to include inflammatory diseases such as Behçet’s disease, gout and idiopathic recurrent pericarditis, all of which have autoinflammatory-mediated mechanisms and a presumed polygenic basis [10]. Identification of the causative genes has also allowed for the confirmation of the clinical diagnosis in familial Mediterranean fever (FMF); tumor necrosis factor receptor-associated periodic syndrome (TRAPS); cryopyrin-associated periodic syndrome (CAPS), which includes familial cold-induced autoinflammatory syndrome (FCAS), Muckle-Wells syndrome (MWS) and chronic infantile neurological cutaneous articular (CINCA) syndrome; mevalonate kinase deficiency (MKD); pyogenic diseases including pyogenic arthritis/pyoderma gangrenosum/acne (PAPA) syndrome, Majeed syndrome (MS) and deficiency of the IL-1 receptor antagonist (DIRA); Blau syndrome (BS); OTULIN-related autoinflammatory syndrome (ORAS) and proteasome-associated autoinflammatory syndromes (PRAAS). A summary of the monogenic ADs is listed in Table 1. Genes associated with these diseases have been sequentially identified since 1997 onwards, and with the exception of MKD, the majority of them encode for proteins involved in the inflammasome architecture or in programmed cell death [11]. In particular, inflammasomes, structured with a central sensor protein, an adaptor protein and CASPASE-1, modulate IL-1β release and work as platforms to protect the human body from the outnumber of pathogenic organisms [12].

## 2. The Classical Hereditary Periodic Fevers

### 2.1. Familial Mediterranean Fever

FMF is the most common among monogenic ADs, characterized by recurrent attacks of fever and self-limited serositis, which may be heralded by abdominal and chest pain. This disease has the highest frequency in the Armenian people, but is highly represented in populations living around the Mediterranean sea, Turks, Arabs and non-Ashkenazi Sephardic Jews. The historical migrations of these ethnic groups to other countries contributed to a consistent spread of FMF throughout the world [13]. The gene responsible for FMF, *MEFV*, contains ten exons and encodes PYRIN, a regulatory protein working in white blood cells, where it recognizes pathogen/danger-associated molecular patterns, being involved in the processing of IL-1ß [14]. This is a monogenic disease identified using an autosomal recessive transmission model, but subsequent studies after the *MEFV* gene was discovered clarified that 30–40% of cases with a clinical diagnosis of FMF carry two pathogenic variants, while 30–40% carry just one single variant. For this reason, it is considered the only recessive condition in humans, known so far, with variants having a dose-dependent gain-of-function effect. Most of the pathogenic *MEFV* mutations occur largely in exon 10, including M694V, M680I, M694I and V726A, leading to gain-of-function mutated PYRINs [15]. The broad spectrum of clinical presentations associated with *MEFV* mutations might reflect the effect of gene dose on the phenotypic expression. Specifically, the role of PYRIN is mediated by Ras homologous guanosine triphosphatase (Rho GTPase), which is targeted to the plasma membrane by geranylgeranyl lipid tails: PYRIN in normal conditions is in an idle state due to the binding of inhibitory proteins to the phosphorylated form of PYRIN; the inactivation of RhoA GTPase by bacterial toxins prevents PYRIN phosphorylation, while inhibitory proteins are released from their binding to PYRIN, leading to PYRIN activation and production of IL-1β (see Figure 1). Accordingly, *MEFV* mutations, particularly those in exon 10, are thought to block PYRIN phosphorylation and thereby activate PYRIN with consequent inflammation [16]. An alkaloid from the *Colchicum autumnale* and *Gloriosa superba* species, colchicine, interacts with tubulin dimers and inhibits microtubule assembly, but also suppresses PYRIN activity through a direct activation of RhoA GTPase [17]. Long periods of uncontrolled inflammation in FMF may induce secondary AA-amyloidosis, which affected up to 60% of patients in the pre-colchicine era and now up to 8.6% of patients. The most severe consequence of amyloidosis is nephrotic syndrome leading to end-stage kidney disease, though it may also involve the cardiovascular, gastrointestinal and central nervous systems. Microalbuminuria may be the first sign of early kidney impairment in FMF, and serum amyloid-A protein is a reliable indicator of overt or subclinical inflammation and indirectly of colchicine treatment compliance [18]. Many organs can be involved during FMF flares and a host of non-classical manifestations have been reported, such as different forms of vasculitis and neurological diseases [19]. There are limited randomized controlled trials assessing interventions for patients with FMF. Since 1972, colchicine (1-to-3 mg/day) is the standard treatment of FMF, preventing acute attacks and amyloidosis or progression to renal failure in most cases, though a significant number of patients are resistant or intolerant to this drug [20]. Anti-IL-1 agents have been successfully used in these cases. The two drugs interfering with IL-1 network are anakinra, an IL-1 receptor antagonist, and canakinumab, a human specific monoclonal anti-IL-1ß antibody. Due to its shorter half-life, anakinra is preferred when the disease is associated with an increased risk of infections (such as dialysis, malignancies or post-transplantation) and in the case of pregnancy, otherwise canakinumab is the preferred drug due to its longer half-life [21]. FMF diagnosis requires the combination of clinical data and ethnicity for both Tel Hashomer and Livneh’s criteria, which are listed in Table 2 [22,23]. Molecular analysis of the *MEFV* gene is useful to confirm the clinical diagnosis of FMF, and it will be important as a future line of research to investigate colchicine-respondent patients with no *MEFV* mutation and disclose whether variants in a different gene might be identified.

### 2.2. Tumor Necrosis Factor Receptor-Associated Periodic Syndrome

TRAPS (or ‘autosomal dominant familial periodic fever’) represents the most common dominant condition among ADs and is related to dysfunction of the tumor necrosis factor (TNF)-α receptor, following specific *TNFRSF1A* mutations [24]. Heterozygous variants related to the extracellular cysteine-rich domains 1 and 2 of the TNF receptor cause TRAPS, and most of the 82 validated pathogenic/likely pathogenic variants in the Infevers database are missense mutations involving a cysteine residue, disrupting the processing and/or the structure of the TNF receptor in leukocytes [25,26]. The disease usually starts in the second infancy with long-lasting febrile attacks which persist for two to three weeks and recur at variable intervals, usually longer than in other ADs, combined with muscle pain and migratory skin rashes [27]. Patients with low-penetrance *TNFRSF1A* mutations have a negative family history and a later clinical onset compared with those carrying frankly pathogenic variants [28]. Amyloidosis is *TNFRSF1A* mutation-dependent and is commonly found as a TRAPS ominous complication [29]. Recurrent self-limited acute pericardial disease is the most common cardiovascular sign in attacks of both TRAPS and FMF [30]. Genetic and clinical heterogeneity make management of TRAPS patients shadowy: corticosteroids can alleviate inflammatory attacks in most of the cases, whereas the TNF inhibitor etanercept is only partially effective in a subgroup of patients [31]. Anti-IL-1 therapies are generally successful in terms of clinical improvement, normalization of inflammatory parameters and even overturning of amyloidosis [32].

### 2.3. Cryopyrin-Associated Periodic Syndrome

CAPS includes three phenotypes, progressively more severe: FCAS, MWS and CINCA syndrome, all representing the same autosomal-dominant disease spectrum, caused by *NLRP3* mutations. The NLRP3 protein, also named cryopyrin, is an intracellular sensor controlling inflammasome-driven innate immunity [33]. There are 80 pathogenic/likely pathogenic validated variants in the Infevers database responsible for the FCAS/MWS/CINCA phenotypes: most of these variants are located in the third exon within the NOD domain of the NLRP3 protein. It is noteworthy that some FCAS/MWS/CINCA patients carry *NLRP3* mosaic variants in a subgroup of leukocytes, making molecular diagnosis difficult unless those variants are specifically sought by next generation sequencing (NGS) analysis. The presence of mosaicism also makes genetic counseling about the recurrence risk more problematic [34]. CAPS patients share the following symptoms: abnormal body temperature or frank fever, flu-like myalgia, cold-induced urticaria-like plaques, headache and ocular inflammatory signs of varying severity, whereas the triad of skin rash, chronic meningopathy and knee arthropathy denote CINCA syndrome [35,36]. The major skeletal abnormality of CINCA patients is caused by the premature aberrant ossification of kneecaps, with hypertrophy of growth plates and structural anomalies involving long bone epiphysis, but skin manifestations consisting of urticaria-like rashes dominate the clinical picture [37]. Endocrine abnormalities may also contribute to stunted growth of CINCA children [38]. Early treatment is crucial to avoid organ damage, however for many years the treatment of this condition has been exclusively supportive, and many different drugs including TNF-inhibitors gave disappointing results [39]. The discovery of *NLRP3* mutations, causing increased IL-1 secretion led to a targeted treatment with IL-1 blockers, changing the natural history of these diseases [40,41]. Both canakinumab, the specific anti-IL-1β antibody, and anakinra, the IL-1 receptor antagonist, have been approved for CAPS [42]. These same drugs are effective in controlling the most severe manifestations of CINCA syndrome affecting the central nervous system symptoms and knees [43,44]. Quality of life in all CAPS patients is significantly affected by persistent fatigue of unknown origin, while the most severe complication is represented by amyloidosis, occurring in about 20% of overlooked cases [45].

### 2.4. Mevalonate Kinase Deficiency

MKD, also known as ‘hyper-IgD syndrome’ due to the common elevation of serum IgD found in these children, is caused by deficient or aberrant activity of mevalonate kinase, the second enzyme in the cholesterol biosynthesis pathway, as a consequence of mutations in the *MVK* gene [46]. This condition is transmitted as a classical autosomal recessive entity, therefore both parents are healthy heterozygous carriers of variants with loss-of-function effect on the enzyme. Usually in patients with hyper-IgD syndrome, the mevalonate kinase activity is higher compared to patients with mevalonic aciduria, and for this reason, urinary excretion of mevalonic acid is increased only during flares. Therefore, urinary excretion of mevalonic acid is decisive for MKD diagnosis along with the identification of pathogenic variants in *MVK* [47]. Many children with MKD have a dramatic history of recurrent fevers, which may be overlooked by the general pediatrician. In ¾ of cases, the disease begins within the first year of life with high fevers recurring every month combined with joint pain, lymphadenopathy, gastrointestinal symptoms, mucosal ulcers and variable skin rashes which recur throughout life, though flares tend to lessen in adulthood [48]. Therapeutic tools for MKD are numerous and differently effective, as non-steroidal anti-inflammatory drugs or biological agents targeting IL-1, such as anakinra (also given ‘on-demand’) and canakinumab have been used; hematopoietic stem cell transplantation has been addressed to the most severe patients or in those with mevalonic aciduria [49]. Table 3 lists the Eurofever/PRINTO classification criteria for FMF, TRAPS, CAPS and MKD [50].

## 3. The Pyogenic Diseases with an Autoinflammatory Basis

Hereditary ADs with pyogenic manifestations have the tendency to develop inflammation with neutrophil-rich sterile exudate, involving bone tissue and skin. Arthritis and cutaneous abscesses with cystic acne and pyoderma gangrenosum characterize PAPA syndrome, with early multifocal osteomyelitis frequently affecting the clavicles, long bones or sternum [51]. MS is characterized by dyserythropoietic anemia and different pictures of neutrophilic dermatosis from palmoplantar pustular rash to Sweet’s syndrome, consisting of painful, edematous and erythematous papules, plaques or nodules along with fever [52]. Osteomyelitis and skin pustulosis with onset during the neonatal period are typical of DIRA, which is caused by uncontrolled IL-1 secretion and for this reason shows a dramatically-full response to anakinra [53]. These three disorders are summarized in Table 4. In particular, PAPA syndrome is caused by dysfunction of the proline-serine-threonine phosphatase-interacting protein 1 (PSTPIP1), which acts as a cytoskeleton-associated adaptor. The differential diagnosis requires to exclude other inflammatory skin and bone disorders, even associated with suppurative hidradenitis and/or psoriatic arthritis [54]. The experience with antagonists of either IL-12/IL-23 or IL-17A pathways (ustekinumab or secukinumab, respectively) is limited, but at least skin manifestations of both MS and DIRA respond sharply to IL-1 blockers [55].

## 4. An Overview of the NFKB-Related Autoinflammatory Disorders

Nuclear factor kappa light-chain enhancer of activated B cells (also known as NFKB) is a family of DNA transcription factors which coordinate many genes involved in infection-related immune responses, having variable effects on cell differentiation and survival [56]. Representative of NFKB-related ADs (recently denominated “relopathies”) is BS, also known as ‘familial juvenile granulomatosis’, caused by dysfunction of the apoptosis-regulating protein NOD2 (or CARD15), crucially working as a sensor for bacterial antigens via NFKB pathway [57]. Heterozygous *NOD2* variants mostly located in the NOD/NACHT domain of the NOD2 protein are responsible for the BS phenotype, and two variants-R334W and R334Q-explain 40–80% of all cases of BS [58]. This syndrome is characterized by refractory granulomatous panuveitis, early onset-deforming boggy polyarthritis and a peculiar ‘dirty’ tan-coloured scaly rash with ichthyosis-like features [59]. BS may also present in a sporadic form as “early-onset sarcoidosis”, in which the histological assessment of involved sites reveals noncaseating granulomas with multinucleated giant cells, autophagy and emperipolesis (i.e., penetration of an intact cell within the cytoplasm of another cell) [60]. Treatment of BS relies on corticosteroids, disease-modifying anti-rheumatic drugs, immunosuppressant drugs, TNF inhibitors or IL-1 antagonists according to the specific manifestations to target [61]. Another condition with upregulated NFKB activity is the deficiency of the IL-36 receptor antagonist, named DITRA, which is caused by homozygous loss-of-function mutations in the *IL36RN* gene, coding for this receptor antagonist that belongs to the IL-1 family. Severe early-onset pustular rashes, sometimes generalized, sometimes limited to the limbs characterize this disorder [62]. Furthermore, the combination of generalized pustular psoriasis, early-onset plaque psoriasis and pityriasis rubra pilaris may be related to sporadic or dominant gain-of-function mutations in the *CARD14* gene, highly expressed in keratinocytes: the disease is called CARD14-mediated psoriasis (also known as CAMPS) and is characterized by NFKB increased signaling [63].

The NFKB pathway is tightly regulated by multiple post-translational mechanisms, including protein ubiquitination. Ubiquitin metabolism is controlled by deubiquitinase enzymes, such as OTULIN and A20: cells mutant for OTULIN or A20 display constitutive upregulation of NFKB pathway [64]. Hypomorphic mutations in the *OTULIN* gene, resulting in elevated NFKB activity, cause ORAS, also named “otulipenia”, which is characterized by inflammatory skin signs with panniculitis starting during the neonatal period and responding to TNF inhibitors [65]. Similarly, heterozygous haploinsufficient mutations in the *A20 (TNFAIP3)* gene cause abnormal ubiquitination patterns, giving rise to aberrant ubiquitination and NFKB upregulation [66]. Table 5 lists the general manifestations at onset of NFKB-diseases in childhood. Recently, using a genotype-driven approach, a new autoinflammatory disorder named VEXAS syndrome has been identified, involving ubiquitination in hematopoietic cells, as a result of somatic mutations in the *UBA1* gene which encodes the ubiquitin activating enzyme E1: the disorder is characterized by bone marrow failure with cytoplasmic vacuoles in blood precursors, skin, joint or lung inflammatory signs [67]. Another rare NFKB-mediated condition is the *NLRP12*-autoinflammatory disorder, also named ‘FCAS2′ due to its similarity with FCAS. This condition is recognizable for the recurrent cold-induced episodes of fever, skin rash, abdominal pain and lymph node enlargement. However, unlike CAPS, it is poorly responsive to anakinra [68].

## 5. Looking through Interferon-Related Autoinflammatory Disorders

IFNs have a variety of antiviral, antitumor and immunoregulatory activities: interferonopathies with autoinflammatory mechanisms have been identified through the recognition of IFN-upregulated genes (or IFN-signature) in the peripheral blood and variable extents of systemic inflammation. The most relevant of these disorders is PRAAS, which is caused by loss-of-function mutations in genes related to proteasome components such as *PSMB8*, the most relevant, but also in *PSMB7, PSMB9, PSMA3* and proteasome assembly factors as *POMP* or *PSMG2* [69]. In particular, the proteasome system is crucial for intracellular extra-lysosomal degradation of proteins, and IFN blockade with Janus kinase inhibitors has been proved to ensure the improvement of clinical manifestations of most patients with interferonopathies [70]. A defect in the degradation of dismissed proteins causes PRAAS, a group of rare diseases in which the accumulation of ubiquitinated waste proteins activates type I IFN signaling with subsequent inflammation: among PRAAS the best-known is the “chronic atypical neutrophilic dermatosis with lipodystrophy and elevated temperature” (or CANDLE) syndrome. Muscle atrophy, contractures of small joints and recurrent fevers are frequently observed in PRAAS in combination with atypical neutrophilic dermatosis ranging from violaceous plaques to erythema nodosum-like panniculitis, histologically resembling a “leukemia cutis”, and usually evolving to localized areas of lipodystrophy: lipodystrophic skin lesions are difficult to recover and give these children a typical appearance [71,72]. *STING*-associated vasculopathy with infancy onset is a rare IFN-mediated disorder characterized by early vasculitis localized to cheeks, ears, nose and fingers (bearing a high risk of gangrene) and chronic interstitial lung disease, for which there is constitutive activation of the *STING* protein, a key activator of type I IFN axis [73]. Finally, Aicardi–Goutières syndrome (AGS) is a disease spectrum caused by upregulated type I IFN production, which shows a noteworthy overlap with transplacental congenital infections involving the central nervous system and skin [74]. This subacute encephalomyelitis is genetically heterogenous and genetic abnormalities induce the loss of enzymatic functions crucial for regulating DNA and RNA metabolism. The 2021 European Alliance of Associations for Rheumatology/American College of Rheumatology published some clues for guiding health care professionals involved in diagnosis and management of these specific disorders, which have no definite cure or treatment so far [75]. Table 6 lists the most relevant features of autoinflammatory interferonopathies at their onset.

## 6. Insights on the Polygenic and Multifactorial Autoinflammatory Disorders in Children

Autoinflammatory diseases of unknown etiology with a presumed either polygenic or multifactorial origin have been recognized in adult people, including Behçet’s disease, idiopathic recurrent pericarditis, crystal-induced arthropathies and adult-onset Still’s disease, but also in children, including systemic juvenile idiopathic arthritis (sJIA), Kawasaki disease (KD) and PFAPA syndrome. These conditions are largely characterized by dysregulation of the innate immune network and upregulation of inflammasome-associated genes [76]. The whole IL-1 cytokine family is abundantly involved in polygenic/multifactorial ADs, and different randomized placebo-controlled clinical trials have confirmed the efficacy of IL-1 inhibitors in their management [77]. Symptoms of such complex heterogeneous diseases, including recurrent fevers, synovitis and serositis, may overlap with monogenic ADs; in addition, some well-known life-threatening complications as macrophage activation syndrome or reactive amyloidosis might occur [78,79,80]. In particular, sJIA, part of the group of childhood arthritides, has peculiar characteristics derived from uncontrolled activation of phagocytes and hypersecretion of both IL-1 and IL-6, differently from other forms of juvenile idiopathic arthritis. Indeed, sJIA is marked by extra-articular signs that include spiking fevers, rash, hepatosplenomegaly, generalized lymphadenopathy and serositis [81]. Table 7 defines sJIA according to the PRINTO classification criteria [82].

Another febrile disorder with a supposed autoinflammatory basis is KD, an acute self-limiting vasculitis of unknown etiology, usually affecting children younger than 5 years, particularly those of Asian descent. This condition has a typical monophasic course characterized by unremitting high fever combined with a constellation of nonspecific skin, orofacial or cervical signs: the risk of coronary artery damage in 1/4 of untreated patients makes KD the most common cause of acquired heart disease for children living in the developed world [83,84]. Despite more than 5 decades of enquiries, the underlying mechanisms provoking coronary artery suffering in KD remain unknown: some scientific gaps include genetic predisposition to KD, dysregulated activation of autoinflammatory pathways and tendency to recur in a minority of cases. A series of studies indicate that the *primum movens* may be an abnormal immune response to different infectious agents, causing both endothelial cell malfunction and turbulent inflammatory cascade in a genetically-predisposed child [85]. Treatment with intravenous immunoglobulin during the first 10 days of disease decreases the risk of developing coronary artery aneurysms by five-fold, but non-responders are those with the higher risk of developing heart complications [86,87]. The multi-systemic inflammatory syndrome seen in children with ‘coronavirus disease 2019′ has been described to partially overlap with KD, but the upregulation of autoinflammation-related genes and hypersecretion of IL-1α, IL-6 and TNF are typical of KD [88]. Early identification of KD patients refractory to immunoglobulin might allow a more intensive treatment to prevent coronary artery abnormalities: for instance, Koné-Paut et al. recently found that intravenous immunoglobulin-resistant KD patients could be successfully treated with IL-1 blockade (i.e., anakinra) to obtain temperature normalization and overall improvement of vasculitis-related manifestations [89].

The most frequent cause of recurrent fevers in children younger than 5 years remains PFAPA syndrome, defined by febrile attacks having “clockwork” periodicity with stereotyped signs affecting the oral cavity and neck alternated by periods of whole well-being: this autoinflammatory disease displays a negative impact on child’s and parents’ quality of life, though its outcome is generally favorable due to remission after an unpredictable period of months or years [90]. There are limited studies focusing on the cyclic nature of PFAPA symptom recurrence, though clock-related genes and their interaction with different immunologic activities have been proven [91]. PFAPA clinical picture overlaps with several other causes of recurring fevers in the pediatric population, such as recurrent tonsillitis, Behçet’s disease and mostly cyclic neutropenia, an ultra-rare hereditary condition diagnosed via demonstration of periodic oscillations of the neutrophil count every 21 days [92]. Adults with de novo PFAPA syndrome or with reappearance of PFAPA symptoms after a first disease resolution have been increasingly reported, but they have a less strict phenotype [93,94]. IL-1-mediated PFAPA pathogenesis suggests that the syndrome could be framed as a rhythmic self-limited dysregulation of innate immunity, disrupting the commensal oral ecosystem and specifically the microbial community in the tonsils [95]. The genetic susceptibility of PFAPA syndrome is yet to prove, although many overlapping symptoms with monogenic hereditary ADs, its dominant recurrency in about 10% of PFAPA patients, hyperexpression of inflammasome-associated genes during febrile flares, and therapeutic efficacy of IL-1 blockers suggest a potential genetic origin [96]. Recently, Sangiorgi et al. studied a small population of familial and sporadic cases of PFAPA-like patients identifying variants in the *ALPK1* gene [97]; a specific missense mutation (T237M) in this same gene has been also related to a new autosomal dominant autoinflammatory condition called ROSAH syndrome, characterized by retinal dystrophy, optic nerve edema, splenomegaly, anhidrosis and migraine [98]. More studies are obviously needed to confirm the role of such rare *ALPK1* variants in larger cohorts of PFAPA patients. Table 8 shows the definition of PFAPA syndrome in childhood according to Marshall’s criteria [99], criteria set off for adult patients [100], and according to Eurofever/PRINTO classification criteria [50].

## 7. Genetic In-Depth Analyses for Autoinflammatory Disorders

Several ADs share many common symptoms and, in spite of well-defined criteria for FMF or other monogenic conditions, sometimes it is tricky to categorize a specific pathologic entity. Since the introduction in the clinical practice of short read sequencing, also commonly known as NGS, it is nowadays practical and convenient to analyze a panel of genes potentially involved in autoinflammatory conditions and test them in every patient arriving in our clinics with the suspicion of AD. Several studies validated the approach of testing different genes at the same time, and the effectiveness of NGS strategy evaluating 32–55 genes has been found superior to the clinical-based gene Sanger sequencing for the genetic diagnosis of ADs in a specific individual [101]. Even when the selection of patients was very stringent, and clinicians involved in patients’ selection were expert in the field, the gene panel analysis was superior in terms of number of “variant of unknown significance” or “pathogenic” that were discovered in respect to the Sanger sequencing of a single gene. Moreover, the cost of reagents, personnel, and time spent sequencing a single gene is perfectly comparable to the cost of sequencing a panel of 10–20 genes. This approach in the long-term has also the advantage of helping to select sporadic patients, and especially families with ADs, for sequencing the whole exome. The ultimate goal would be discover new variants in new genes in all those patients mutation-negative for genes already found mutated, as was demonstrated for *ALPK1* and ROSAH syndrome.

## 8. Conclusive Remarks

Great progress has been achieved in the recognition of non-infectious, non-rheumatic or non-tumoral causes of recurrent fever, and dysregulated inflammasome activity caused by mutations in genes coding for inflammasome components and/or their interaction partners leads to protean inflammatory scenarios associated with fever in children [102,103,104]. This family of conditions with seemingly unprovoked attacks of systemic inflammation with neither infectious antigens, nor autoantibodies and autoreactive T cells, is expanding. With the aim of improving the diagnosis of ADs at a genetic level in children under the age of 10, some diagnostic scores have been proposed [105,106], and many scoring systems have been spread for assessing disease activity or establishing therapeutic efficacy during follow-up [107], though real-life studies for evaluating outcomes of the various ADs are still in progress.

The definition of ADs should be revised, reconsidering these diseases as depicted by abnormal inflammation patterns mediated by the innate immune system. Such definition should include not only the classical monogenic forms, but also a broader range of common diseases with a presumed polygenic or multifactorial basis, for which the contribution of deregulated inflammasomes to pathogenesis is corroborated by the discovery of effective IL-1 blockade at a clinical level. As new studies will continue to unravel the chameleonic innate immune system’s arsenal of sensory proteins and the family of ADs further expands, it is likely that the different pathways will be better clarified and a more tailored employment of novel anti-inflammatory agents combined with host-directed therapies will give a definite chance for a personalized management of both hereditary and non-hereditary ADs.


**Keypoints**
-Dysregulated inflammasome activity and oversecretion of interleukin-1 are the ‘*incipit*’ of hereditary autoinflammatory diseases, caused by mutations in genes coding for inflammasome pieces and giving rise to protean scenarios in different pediatric settings.-Familial Mediterranean fever, tumor necrosis factor receptor-associated periodic syndrome, cryopyrin-associated periodic syndrome, mevalonate kinase deficiency, but also some idiopathic pyogenic diseases, granulomatous diseases and defects of the ubiquitin-proteasome pathway are monogenic autoinflammatory diseases with a distinct molecular pathogenesis.-In each of these diseases, inflammatory episodes recur over time, often with minimal evidence of a provocative event, and patients do not display high-titer autoantibodies or antigen-specific T cells that might serve as mediators.-A further cluster of diseases of unknown etiology with either a presumed polygenic or multifactorial origin, as systemic juvenile idiopathic arthritis, Kawasaki disease and periodic fever/aphthous stomatitis/pharyngitis/cervical adenopathy syndrome, is characterized by dysregulation of innate immune responses and overexpression of inflammasome-associated genes.-Despite the recent advances in genetic testing, the diagnosis of autoinflammatory diseases is based on a thorough knowledge of the clinical phenotype, and this knowledge should assist when children present unexplained episodes of fever or inflammation.


## Figures and Tables

**Figure 1 cells-11-02231-f001:**
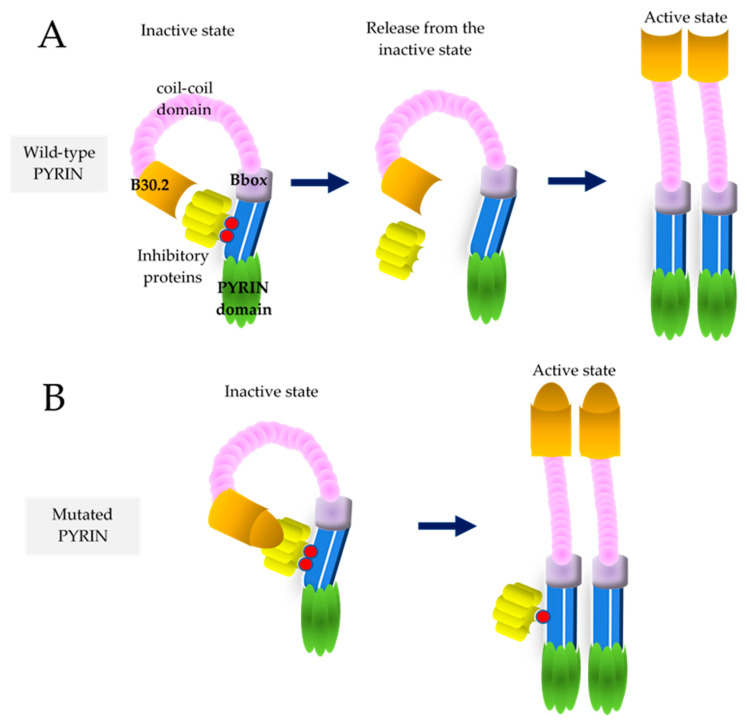
Schematic representation of PYRIN activation in wild-type and mutated conditions. (**A**) In wild-type conditions phosphorylated PYRIN is kept inactive by inhibitory proteins; once phosphorylation is lost, i.e., due to bacterial invasion, the link between PYRIN and inhibitory proteins fails and PYRIN inflammasome becomes fully active. (**B**) Mutated PYRIN has an unstable link with the inhibitory proteins and the PYRIN inflammasome tends to activation.

**Table 1 cells-11-02231-t001:** Brief summary of the hereditary monogenic autoinflammatory diseases.

	*Gene* Locus	Protein	Inheritance	Main Manifestations and Complications	Available Treatments
**FMF**	*MEFV* 16p13.3	PYRIN or marenostrin	AR	serositis, limb pain or transient arthritis, erysipelas-like eruption on the legs, nonspecific skin manifestations (like urticaria, angioedema, erythema nodosum, vasculitis), risk of amyloidosis	Colchicine, canakinumab, anakinra
**TRAPS**	*TNFRSF1A* 12p13	TNFRSF1A, TNF receptor	AD	severe migrating muscle pain, arthralgia or arthritis, serositis, painful orbital edema, painful conjunctivitis, risk of amyloidosis	Canakinumab, anakinra, corticosteroids
**FCAS**	*NLRP3* 1q44	CRYOPYRIN	AD	cold-induced urticaria-like rashes, conjunctivitis, arthralgia	Anakinra, rilonacept, canakinumab
**MWS**				cold-induced urticaria-like rash, conjunctivitis, episcleritis, arthralgia, neurosensorial deafness, risk of amyloidosis	
**CINCA**				migrating non-itchy urticaria-like rash, uveitis, papilledema, deforming arthritis involving large joints, aseptic chronic meningopathy, retinal dystrophy, neurosensorial deafness, risk of amyloidosis	
**MKD**	*MVK* 12q24	MEVALONATE KINASE	AR	fatigue, painful generalized lymph node enlargement, vomiting, diarrhea, abdominal pain, arthralgia, skin rashes of varying severity, oral and/or genital aphthosis, splenomegaly during flares	Anti-inflammatory drugs, corticosteroids, anakinra ‘on demand’, canakinumab
**PAPA**	*PSTPIP1* 15q24–25	PSTPIP1 (proline-serine- threonine phosphatase interacting protein 1)	AD	sterile pyogenic arthritis, pyoderma gangrenosum, severe acne, skin abscesses, recurrent non-healing sterile ulcers	Corticosteroids, infliximab, anakinra, immunosuppressive agents
**MS**	*LPIN2* 18p11.31	LIPIN2 (phosphatidate phosphatase)	AR	recurrent multifocal osteomyelitis, neutrophilic dermatosis, dyserythropoietic anemia	Corticosteroids, bisphosphonates, TNF-α inhibitors, IL-1 antagonists (anakinra)
**DIRA**	*IL1RN* 2q14.1	IL1RN (interleukin-1 receptor antagonist)	AR	sterile multifocal osteomyelitis starting in the neonatal period, skin pustulosis, osteitis	Anakinra
**BS**	*NOD2 (CARD15)* 16q12.1–13	NOD2 (nucleotide binding oligomerization domain containing 2)	AD	non-erosive granulomatous polyarthritis (‘boggy synovitis’ with painless effusion and cyst-like swelling of joints), granulomatous panuveitis, skin granulomatous rash	Corticosteroids, TNF-α inhibitors (infliximab), IL-1 antagonists, JAK inhibitors (tofacitinib)
**DITRA**	*IL36RN* 2q14.1	IL36RN (interleukin-36 receptor antagonist)	AR	severe pustular psoriasis (generalized or limited to the distal part of limbs)	TNF-α inhibitors (adalimumab), IL-12/23 antagonists, IL-17 antagonists
**CAMPS**	*CARD14* 17q25.3	CARD14 (caspase recruitment domain- containing protein 14)	AD	psoriasis in a wide range of phenotypes	Methotrexate, corticosteroids, cyclosporine, phototherapy, acitretin, vitamin D analogs, TNF-α inhibitors, IL-12/23 antagonists, IL-17 antagonists
**ORAS**	*OTULIN* 5p15.2	OTULIN (deubiquitinase)	AR	fever starting in the neonatal period, neutrophilic dermatosis associated with panniculitis, growth retardation	TNF-α inhibitors
**HA20**	*TNFAIP3* 6q23.3	TNFAIP3 (tumor necrosis factor alpha-induced protein 3, A20)	AD	recurrent mucosal ulcerations of the oral cavity, gastrointestinal tube and urogenital tract, skin rashes, polyarthritis, uveitis, vasculitides, recurrent fevers, association with different autoimmune disorders (systemic lupus erythematosus, psoriatic arthritis, juvenile idiopathic arthritis, autoimmune hepatitis and Hashimoto thyroiditis)	TNF-α inhibitors, colchicine
**FCAS2**	*NLRP12* 19q13.42	NLRP12 (nucleotide- binding oligomerization domain, leucine rich repeat and pyrin domain containing 12)	AD	cold-induced rashes, joint pain, abdominal pain, sensorineural deafness, headache	Anakinra, TNF-α inhibitors, IL-6 antagonists (tocilizumab)
**PRAAS**	*PSMB8*	PSMB8 (proteasome 20s subunit beta 8)	AR	chronic atypical neutrophilic dermatosis, lipodystrophy, erythema nodosum-like panniculitis, abnormal growth of lips, muscular weakness and atrophy, severe joint contractures, basal ganglia calcifications, ear and nose chondritis, aseptic meningitis, conjunctivitis, hepatosplenomegaly, lymph node enlargement, arthralgia	Corticosteroids, immunosuppressive agents, anakinra, IL-6 antagonists (tocilizumab), TNF-α inhibitors, dapsone, JAK inhibitors (baricitinib)
**SAVI**	*STING1* (TMEM173) 5q31.2	STING1 (stimulator of interferon genes protein 1)	AD	vasculopathy causing severe skin lesions on face, ears, nose and digits, resulting in ulcerations, necrosis or amputations, chronic interstitial lung disease	JAK inhibitors (ruxolitinib)
**AGS**	*TREX1,* *RNASEH2B,* *RNASEH2C,* *RNASEH2A,* *SAMHD1,* *ADAR,* *IFIH1* 3p21.31, 13q14.3, 11q13.1, 19p13.13, 20q11.23, 1q21.3, 2q24.2	Enzymes involved in the duplication, repair and recombination of nucleic acids	AR (AD for *IFIH1*)	leukoencephalopathy (mimicking transplacental infections), calcifications in cerebral and basal ganglia, dystonia, microcephaly, cognitive impairment, abnormal eye movements, glaucoma, livedo reticularis, digital chilblain lesions on hands and feet, hepatosplenomegaly, jaundice, silent positivity of autoantibodies	No cure is available, corticosteroids and intravenous immunoglobulin may control systemic and organ inflammation

FMF: familial Mediterranean fever; TRAPS: tumor necrosis factor receptor-associated periodic syndrome (autosomal dominant familial periodic fever); FCAS: familial cold-induced autoinflammatory syndrome; MWS: Muckle-Wells syndrome; CINCA s.: chronic infantile neurologic cutaneous articular syndrome; MKD: mevalonate kinase deficiency (hyper-IgD syndrome); PAPA s.: pyogenic arthritis, pyoderma gangrenosum and acne (PAPA) syndrome; MS: Majeed syndrome; DIRA: deficiency of IL-1 receptor antagonist; BS: Blau syndrome; DITRA: deficiency of the interleukin-36 receptor antagonist; CAMPS: CARD14-mediated psoriasis; HA20: haploinsufficiency of A20; ORAS: OTULIN-related autoinflammatory syndrome; FCAS2: familial cold autoinflammatory syndrome 2 (*NLRP12*-associated autoinflammatory disorder); PRAAS: proteasome-associated autoinflammatory syndrome; SAVI: *STING*-associated vasculopathy with onset in infancy; AGS: Aicardi-Goutières syndrome. AR: autosomic recessive; AD: autosomic dominant; TNF: tumor necrosis factor; IL-1: interleukin-1; JAK: Janus kinase.

**Table 2 cells-11-02231-t002:** Classification criteria for the clinical diagnosis of familial Mediterranean fever (FMF).

Tel Hashomer Criteria	Livneh’s Criteria
**Major**	**Major**
Recurrent fevers + peritonitis/pleurisy/serositis	Typical attack of peritonitis
AA-amyloidosis	Typical attack of unilateral pleuritic or pericarditis
Favourable response to prophylaxis with colchicine	Typical attack of monoarthritis
**Minor**	Fever (rectal temperature of 38 °C or higher) alone
Recurrent fevers	**Minor**
Erysipelas-like erythema	Incomplete attack involving the abdomen
Family history of FMF in a first-degree relative	Incomplete attack involving the chest
	Incomplete attack involving one large joint
	Exertional leg pain
	Favourable response to prophylaxis with colchicine
	**Supportive**
	Family history of familial Mediterranean fever
	Typical ethnic origin (Armenian, Turkish, Arabian, Sephardic Jew)
	Age less than 20 years at disease onset
	Severity of attacks requiring bed rest
	Spontaneous remission of attacks
	Symptom-free intervals between attacks
	Transient increase of inflammatory parameters during attacks
	Episodic proteinuria or hematuria
	Surgical removal of a “white” appendix
	Consanguinity of parents

Diagnosis of FMF is made when 2 major criteria or 1 major and 2 minor criteria are satisfied (according to the Tel Hashomer criteria); diagnosis requires ≥ 1 major criteria, or ≥2 minor criteria, or 1 minor criterion plus ≥ 5 *supportive criteria* or 1 minor criterion plus ≥ 4 of the “first” five *supportive criteria* (according to Livneh’s criteria). ***Note:*** “Incomplete” attacks are defined as painful and recurrent flares that differ from typical attacks in 1 or 2 features, as follows: (a) normal temperature or lower than 38 °C; (b) attacks longer than 1 week or shorter than 6 h; (c) no signs of peritonitis recorded during acute abdominal attacks.

**Table 3 cells-11-02231-t003:** Eurofever/PRINTO classification criteria for the main four hereditary periodic fevers, published in 2019 for identifying patients with fevers recurring in a period of at least 6 months (combined with elevation of inflammatory parameters), who can be recruited for experimental studies.

Familial Mediterranean Fever (FMF)	Autosomal Dominant Familial Periodic Fever (Tumor Necrosis Factor Receptor-Associated Periodic Syndrome, TRAPS)	Cryopyrin-Associated Periodic Syndrome (CAPS)	Mevalonate Kinase Deficiency (MKD, Hyper-IgD Syndrome)
Presence of confirmatory *MEFV* genotype and at least 1 among: - Attacks lasting 1–3 days - Arthritis - Chest pain - Abdominal pain or Presence of a non-confirmatory *MEFV* genotype and at least 2 among: - Attacks lasting 1–3 days - Arthritis - Chest pain - Abdominal pain	Presence of confirmatory *TNFRSF1A* genotype and at least 1 among: - Attacks lasting ≥7 days - Myalgia - Migratory skin rash - Positive family history or Presence of a non-confirmatory *TNFRSF1A* genotype and at least 2 among: - Attacks lasting ≥7 days - Myalgia - Migratory skin rash - Positive family history	Presence of confirmatory *NLRP3* genotype and at least 1 among: - Urticaria-like rash - Eye inflammation - Sensorineural hearing loss or Presence of a non-confirmatory *NLRP3* genotype and at least 2 among: - Urticaria-like rash - Eye inflammation - Sensorineural hearing loss	Presence of confirmatory *MVK* genotype and at least 1 among: - Gastrointestinal symptoms - Cervical lymphadenopathy - Aphthous stomatitis

**Table 4 cells-11-02231-t004:** Onset and general manifestations of the “pyogenic” autoinflammatory disorders in childhood.

	*Onset*	*Clinical Signs Observed in Children at Onset*
**Pyogenic arthritis/pyoderma gangrenosum/acne** (PAPA) **syndrome**	First infancy	Skin ulcerations, pyoderma gangrenosum often associated with cystic acne, sterile pyogenic oligoarthritides
**Majeed syndrome** (MS)	First 2 years of life	Diffuse neutrophilic dermatosis, chronic non-bacterial osteomyelitis, dyserythropoietic anemia
**Deficiency of the interleukin-1 receptor antagonist** (DIRA)	Neonatal period	Pustular rash, nail abnormalities, ichthyosis-like changes of the skin, multifocal osteomyelitis, multi-organ failure

**Table 5 cells-11-02231-t005:** Onset and general manifestations of NFKB-diseases (‘relopathies’) in childhood.

	*Onset*	*Clinical signs observed in children at onset*
**Blau syndrome** (BS)	First infancy	Brown-coloured scaly and ichthyosis-like or lichenoid rashes, recurrent polyarthritis, granulomatous uveitis (anterior, posterior or intermediate), risk of ocular sequelae (synechiae, cataracts, band keratopathy)
**Deficiency of the interleukin-36 receptor antagonist** (DITRA)	Variable (many cases may start in the first infancy)	Generalized severe pustular psoriasis, acute generalized exanthematous pustulosis, pustulosis of palms and soles, disseminated subcorneal pustules, Hallopeau’s acrodermatitis continua, recurrent fevers
**CARD14-mediated psoriasis** (CAMPS)	Variable (many cases may start in the first infancy)	Plaque psoriasis, pityriasis rubra pilaris, pustular psoriasis, joint pain, recurrent fevers
**OTULIN-related autoinflammatory syndrome** (ORAS)	First infancy	Erythematous skin rash with nodules, joint and abdominal pain, diarrhea, lymph node enlargement, stunted growth, recurrent fevers
**Haploinsufficiency of A20** (HA20)	First or second decade	Early-onset Behçet’s-like disease signs as aphthous stomatitis, oral and genital ulcers and/or uveitis combined with diffuse lymphadenopathy, arthritis, recurrent fevers

**Table 6 cells-11-02231-t006:** Onset and general manifestations of interferon-related autoinflammatory disorders in childhood.

	*Onset*	*Clinical Signs Observed in Children at Onset*
**Proteasome-associated autoinflammatory syndromes** (PRAAS)	Early childhood	Recurrent fevers, nodular skin rashes and annular violaceous plaques evolving to panniculitis-induced lipodystrophy (loss of adipose tissue), large nose, lips and ears, eyelid swelling, muscle weakness and atrophy, joint contractures, disproportionately long and thick fingers, hepatosplenomegaly, basal ganglia calcifications
***STING*-associated vasculopathy** **with infancy onset** (SAVI)	Neonatal period	Skin vasculitis with violaceous scaling or pernio-like lesions on fingers, toes or nose, usually exacerbated by cold exposure, that progress to ulcerations of extremities or to autoamputation phenomena, chronic interstitial lung disease
**Aicardi-Goutières syndrome** (AGS)	Neonatal or prenatal period	Subacute encephalopathy, cerebral and basal ganglia calcifications, spasticity, dystonia, microcephaly, eye abnormalities, livedo reticularis, digital vasculitis with chilblain lesions on hands and feet, hepatosplenomegaly, silent positivity of autoantibodies

**Table 7 cells-11-02231-t007:** Classification criteria of systemic juvenile idiopathic arthritis (according to PRINTO organization): fever has to be associated with 2 major criteria or with 1 major criterion and 2 minor criteria, after exclusion of infectious, neoplastic, autoimmune and hereditary autoinflammatory diseases.

*Cardinal Sign*	*Major Criteria*	*Minor Criteria*
Fever of unknown origin that is documented to be daily (until 39 °C once a day with intermittent course) for at least 3 consecutive days and reoccurring over an observation period of at least two weeks	(a) evanescent nonfixed erythematous rash (b) arthritis	(1) generalized lymph node enlargement and/or hepatomegaly and/or splenomegaly (2) serositis (3) arthralgia lasting 2 weeks or longer (in the absence of arthritis) (4) leukocytosis (≥15,000/mm^3^) with neutrophilia

**Table 8 cells-11-02231-t008:** Definition of the periodic fever/aphthous stomatitis/pharyngitis/cervical adenopathy (PFAPA) syndrome in children, in adults and according to the Eurofever/PRINTO classification criteria.

**PFAPA syndrome in children**	Periodically recurring high fevers (with “clockwork” periodism at intervals of 4–6 weeks) + Onset before 5 years + Child’s complete wellness between attacks (with normal growth and no sequelae)	At least 1 among: (a) aphthous stomatitis (b) pharyngitis (c) cervical lymphadenitis Absence of respiratory infection-related symptoms	To rule out: - Cyclic neutropenia - Recurrent upper respiratory infections - Monogenic hereditary sautoinflammatory disorders
**PFAPA syndrome in adults** (at least 16-year-old)	Recurrent fevers + Increased inflammatory parameters during febrile attacks + Symptom-free intervals	At least 1 between: - pharyngitis - cervical lymphadenitis	To rule out: - Infections - Autoimmune disorders - Monogenic hereditary autoinflammatory disorders
Eurofever/PRINTO classification criteria for **PFAPA syndrome**	At least 7 out of the following 8 signs (either positive [from *a* to *d*] or negative [from *e* to *h*]):	(a) pharyngotonsillitis (b) febrile flares lasting 3–6 days (c) cervical lymphadenitis (d) periodic recurrence of flares	(e) absence of diarrhea (f) absence of chest pain (g) absence of skin rash (h) absence of arthritis

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
