# Peer review of "The Clinical Chameleon of Autoinflammatory Diseases in Children"

_cells, 2022, doi:10.3390/cells11142231_

Round 1
Reviewer 1 Report
Authors have performed an interesting and valuable review of autoinflammatory diseases. However, your manuscript could be developed a bit further.
Comments.
1. Authors presented PYRIN associated mechanism. Those matters may seem a bit complicated. Additional graphical presentation might be informative.
2. Chapter2 deals with multiple diseases. It would be easier to understand if it were divided into paragraphs with headings.
3. I think Chapter 6 is a bit abstract. I think it would be better to be more specific in explaining the results and conclusions of the literature 101.
Author Response
Dear Editor-in-Chief of Cells,
we would like to thank sincerely you and both reviewers for the positive comments and suggestions given to ameliorate the overall quality of our paper entitled “The Clinical Chameleon of Autoinflammatory Diseases in Children”.
This is a point-by-point list of changes made according to 1st reviewer’s suggestions.
Authors have performed an interesting and valuable review of autoinflammatory diseases. However, your manuscript could be developed a bit further. Comments: the authors presented PYRIN associated mechanism, which may seem a bit complicated. Additional graphical presentation might be informative.
We have added a figure depicting the PYRIN pathogenetic mechanism active in familial Mediterranean fever (FMF), as suggested.
Chapter 2 deals with multiple diseases. It would be easier to understand if divided into paragraphs with headings.
We have split the second chapter in 4 periods (and 4 subheadings) dedicated to the 4 diseases presented in the group of classical hereditary periodic fevers.
I think Chapter 6 is a bit abstract. I think it would be better to be more specific in explaining the results and conclusions of the literature 101.
We have tried to organize the sixth chapter dedicated to the pediatric polygenic and multifactorial autoinflammatory disorders in a more schematic fashion, paying more attention to clinical details. We have also explained the content of the #ref. no. 101, writing that the effectiveness of the NGS strategy evaluating 32-55 genes has been found superior to the clinical-based gene Sanger sequencing for the genetic diagnosis of ADs.
We sincerely hope that all amendments and changes made in our paper might be judged favorably.
We hope to receive as soon as possible the final decision related to the spread of this paper to Cells readers.
Yours sincerely,
Eugenio Sangiorgi & Donato Rigante

Reviewer 2 Report
The authors reviewed the clinical features of childhood autoinflammatory diseases. They summarized this data with appropriate tables. However, I do have a few criticisms. As it is known, diagnostic criteria and classification criteria are different things. Authors stated that “Molecular analysis of the MEFV gene is necessary to confirm the clinical diagnosis of FMF…” I think this sentence warrants reconsideration in the context of its validity. As stated in the text, Eurofever/PRINTO classification criteria should not be defined as diagnostic. The diagnosis of FMF is still based on clinical features.
The authors defined vitiligo as a skin manifestation of FMF. Except for a few cases, no study in the literature confirms this. It would be good to ignore this information.
Author Response
Dear Editor-in-Chief of Cells,
we would like to thank sincerely you and both reviewers for the positive comments and suggestions given to ameliorate the overall quality of our paper, entitled “The Clinical Chameleon of Autoinflammatory Diseases in Children”.
This is a point-by-point list of changes made according to 2nd reviewer’s suggestions.
The authors reviewed the clinical features of childhood autoinflammatory diseases. They summarized data with appropriate tables. However, I do have a few criticisms. As it is known, diagnostic criteria and classification criteria are different things. Authors stated that “Molecular analysis of the MEFV gene is necessary to confirm the clinical diagnosis of FMF…” I think this sentence warrants reconsideration in the context of its validity.
Of course, most criteria available for autoinflammatory disorders are classification criteria. We have changed the adjective in the suggested sentence, clarifying that molecular analysis of the MEFV gene is simply “useful” to confirm the clinical diagnosis of FMF.
As stated in the text, Eurofever/PRINTO classification criteria should not be defined as diagnostic. The diagnosis of FMF is still based on clinical features.
Of course, we have written that Eurofever/PRINTO classification criteria should not be defined as diagnostic criteria, due to the fact that they are crucial for identifying patients with fevers recurring in a period of at least 6 months who could be recruited for experimental studies. Wherever cited in the text, the Eurofever/PRINTO criteria are defined as “classification criteria”. We agree with your consideration that diagnosis of FMF is based on the clinical stigmata of the disease.
The authors defined vitiligo as a skin manifestation of FMF. Except for a few cases, no study in the literature confirms this. It would be good to ignore this information.
Thank you for this suggestion: we have omitted “vitiligo” among the peculiar skin manifestations occurring in FMF (see Table 1).
We sincerely hope that all amendments and changes ade in our paper might be judged favorably.
We hope to receive as soon as possible the final decision related to the spread of this paper to Cells readers.
Yours sincerely,
Eugenio Sangiorgi & Donato Rigante

Round 2
Reviewer 1 Report
Thank you for your comments.
The additions and corrections to the text and figures make this a clear and attractive review.